# Vitamin D in Cancer Prevention and Treatment: A Review of Epidemiological, Preclinical, and Cellular Studies

**DOI:** 10.3390/cancers16183211

**Published:** 2024-09-20

**Authors:** Siva Dallavalasa, SubbaRao V. Tulimilli, Vidya G. Bettada, Medha Karnik, Chinnappa A. Uthaiah, Preethi G. Anantharaju, Suma M. Nataraj, Rajalakshmi Ramashetty, Olga A. Sukocheva, Edmund Tse, Paramahans V. Salimath, SubbaRao V. Madhunapantula

**Affiliations:** 1Center of Excellence in Molecular Biology and Regenerative Medicine (CEMR) Laboratory (DST-FIST Supported Center and ICMR Collaborating Center of Excellence—ICMR-CCoE), Department of Biochemistry (DST-FIST Supported Department), JSS Medical College, JSS Academy of Higher Education & Research (JSS AHER), Mysuru 570015, Karnataka, India; sivadallavalasa@jssuni.edu.in (S.D.); tulimillivenkatasubbarao@jssuni.edu.in (S.V.T.); vidyabg@jssuni.edu.in (V.G.B.); medhakarniksr@jssuni.edu.in (M.K.); auchinnappa16@gmail.com (C.A.U.); preethiganantharaju@jssuni.edu.in (P.G.A.); mnsuma@jssuni.edu.in (S.M.N.); 2Department of Physiology, JSS Medical College, JSS Academy of Higher Education & Research (JSS AHER), Mysuru 570015, Karnataka, India; rajalakshmir@jssuni.edu.in; 3Department of Hepatology, Royal Adelaide Hospital, Port Rd., Adelaide, SA 5000, Australia; edmund.tse@sa.gov.au; 4JSS Academy of Higher Education & Research (JSS AHER), Mysuru 570015, Karnataka, India; paramahans1954@gmail.com; 5Special Interest Group in Cancer Biology and Cancer Stem Cells (SIG-CBCSC), JSS Medical College, JSS Academy of Higher Education & Research (JSS AHER), Mysuru 570015, Karnataka, India

**Keywords:** vitamin D, cancer, vitamin D receptor, VDR polymorphism, vitamin D analogs, apoptosis, metastasis, cell cycle

## Abstract

**Simple Summary:**

Inhibition of human cancers has previously been linked to the administration of vitamin D. Studies have shown that increased cancer incidence is associated with decreased vitamin D. The anticancer activity of vitamin D has been confirmed by several in vitro and in vivo studies. Vitamin D inhibits the growth of cancer cells by (a) the induction of apoptosis, (b) decreasing the metastatic spread, (c) arresting the cells at the G0/G1 (or) G2/M phase in the cell cycle, and (d) downregulating proliferation signals. Supplementation of vitamin D slows down the growth of xenografted tumors in mice. Hence, vitamin D could be considered a potential cancer chemotherapeutic agent.

**Abstract:**

Background: Inhibition of human carcinomas has previously been linked to vitamin D due to its effects on cancer cell proliferation, migration, angiogenesis, and apoptosis induction. The anticancer activity of vitamin D has been confirmed by several studies, which have shown that increased cancer incidence is associated with decreased vitamin D and that dietary supplementation of vitamin D slows down the growth of xenografted tumors in mice. Vitamin D inhibits the growth of cancer cells by the induction of apoptosis as well as by arresting the cells at the G0/G1 (or) G2/M phase of the cell cycle. Aim and Key Scientific Concepts of the Review: The purpose of this article is to thoroughly review the existing information and discuss and debate to conclude whether vitamin D could be used as an agent to prevent/treat cancers. The existing empirical data have demonstrated that vitamin D can also work in the absence of vitamin D receptors (VDRs), indicating the presence of multiple mechanisms of action for this sunshine vitamin. Polymorphism in the VDR is known to play a key role in tumor cell metastasis and drug resistance. Although there is evidence that vitamin D has both therapeutic and cancer-preventive properties, numerous uncertainties and concerns regarding its use in cancer treatment still exist. These include (a) increased calcium levels in individuals receiving therapeutic doses of vitamin D to suppress the growth of cancer cells; (b) hyperglycemia induction in certain vitamin D-treated study participants; (c) a dearth of evidence showing preventive or therapeutic benefits of cancer in clinical trials; (d) very weak support from proof-of-principle studies; and (e) the inability of vitamin D alone to treat advanced cancers. Addressing these concerns, more potent and less toxic vitamin D analogs have been created, and these are presently undergoing clinical trial evaluation. To provide key information regarding the functions of vitamin D and VDRs, this review provided details of significant advancements in the functional analysis of vitamin D and its analogs and VDR polymorphisms associated with cancers.

## 1. Introduction

Vitamin D, also called the “sunshine” vitamin, is formed when skin is exposed to sunlight. It is well known that vitamin D keeps the serum calcium levels within the physiological range, i.e., 8.5 to 10.2 mg/dL, and preserves bone health [1]. Any concentration of vitamin D less than 30 ng/mL of blood is considered vitamin D insufficiency. To maintain vitamin D levels within the normal range (30–50 ng/mL blood), the Endocrine Society recommends 400–1000 international units (IU; 1 IU = 0.025 µg of vitamin D) every day for infants under the age of one year, 600–1000 IU for children and adolescents, and 1500–2000 IU for adults [2,3]. Earlier studies have demonstrated that vitamin D deficiency is a global health concern [4]. Recent findings in public health research have also demonstrated that one of the most prevalent conditions seen in cancer patients is vitamin D insufficiency [5]. Furthermore, meta-analyses of randomized trials revealed a robust correlation between vitamin D sufficiency and a lower cancer death rate [6]. Vitamin D supplementation, however, does not appear to have any effect on tumor development, cancer-induced mortality, and tumor cells’ susceptibility to radiation. Additionally, several studies evaluating the impact of vitamin D supplementation on cancer incidence and death have yielded poor results, suggesting that more studies are necessary to validate vitamin D’s potential as a cancer preventive and therapeutic [7]. Due to contradictory findings about vitamin D’s role in cancer prevention and therapy [8], an in-depth review is necessary to evaluate vitamin D’s uses and functions. Although it has been shown that vitamin D can inhibit the development of cancer even in cell lines or tissues that express a mutant version of the VDR, additional investigations are needed to determine whether or not VDR expression is a prerequisite for vitamin D to have anticancer activity [9]. Recent studies have demonstrated the prolonged antiproliferative potential and increased biological activity of vitamin D analogs [10]. Therefore, this review also covers recent research on vitamin D analogs.

### 1.1. VDR Structure and Function: A Short Overview

The gene encoding for a VDR is located on the long arm of chromosome 12. The VDR gene is made up of a promoter region, regulatory regions 1a–1f, and exons 2–9. The VDR protein is made up of six domains, viz., A–F (Figure 1) [11]. The nuclear localization region (represented in RED) of the VDR protein guides the receptor into the nucleus [12,13,14,15]. When calcitriol binds to the hormone-binding domain (represented in GREEN color), PKC phosphorylates serine 51 in the VDR DNA-binding domain (represented in YELLOW color), and CKII phosphorylates serine 208 in the hinge region [16,17]. The phosphorylated VDR complex can dimerize with RXR via dimerization domains (represented in BLUE color) and form the calcitriol–VDR–RXR complex. This tripartite complex binds to the VDRE and influences the expression of target genes. The conformational changes in VDR also influence gene expression linked to the separation of the co-repressor SMRT. The interaction of the VDR with the activation function 2 (AF2) transactivation domain (represented in LIGHT GREY color) and stimulatory coactivators (such as steroid receptor coactivators (SRCs)) is made possible by the dissociation of SMRT. The nuclear coactivator-62-kDa-Ski-interacting protein (NCoA62–SKIP) complex, in conjunction with VDR-interacting proteins, enhances the transcriptional activation of VDR target genes [11].

### 1.2. Vitamin D Metabolism and Mechanism: Short Overview

The physiologically active form of vitamin D, calcitriol (1α,25(OH)2D3), is synthesized in a well-controlled multistep process [18]. The two main isoforms of vitamin D, vitamin D2 (ergocalciferol) and vitamin D3 (cholecalciferol), are produced from ergosterol by UV-B radiation. In humans, UV-B radiation can also produce 7-dehydrocholesterol in the epidermis of skin [19,20]. The prohormone vitamin D binds to its nuclear receptors to regulate a variety of physiological processes after being metabolized into a physiologically active substance [9]. In the blood, vitamin D is circulated in the form of 25-hydroxycholecalciferol [25(OH)D3, calcidiol or circulating vitamin D], which is produced by the metabolism of cholecalciferol in the liver by 25-hydroxylase. This hydroxylase is encoded by the CYP27A1 gene [21,22]. The physiologically active 1α,25 hydroxycholecalciferol [1,25(OH)2D3, calcitriol or active form of vitamin D] is produced in the kidney by 25-hydroxyvitamin D3-1α-hydroxylase, which is encoded by the CYP27B1 gene [21,22]. Calcitriol enters the bloodstream, where it binds to the vitamin D-binding protein (VDBP) and travels to the kidney, bone, and gut, among other target organs, to control the uptake, mobilization, and, finally, reabsorption of calcium and phosphate [19]. The hydroxylation at position 24 by the cytochrome P-450 enzyme generates 24,25(OH)2 D3 and 1α,24,25(OH)2 D3 [11]. 24-hydroxylase is encoded by the CYP24A1 gene. An increase in 24,25(OH)2 D3 causes the production of 1α,25(OH)2 D3, which in turn regulates the levels of calcitriol. Increases in Ca2+, inorganic phosphate, and calcitriol itself may inhibit 1α,25(OH)2D3 production [11,23,24,25]. Parathyroid hormone (PTH) induces CYP27B1 expression, while 1α,25(OH)2D3 represses it [23,26]. In contrast to CYP27B1, 1α,25(OH)2D3 substantially induces the expression of CYP24A [27] (Figure 2).

Calcitriol binds to either membrane-bound (non-genomic pathway) or cytosolic (genomic pathway) VDRs in target tissues, thereby inducing the expression of downstream target genes that are involved in controlling several biological processes. The VDR belongs to the family of ligand-activated transcription factors that are nuclear receptors [9,28,29,30]. In the genomic pathway, calcitriol binds with the cytosolic VDR to create a complex that stimulates protein kinase C (PKC) and casein kinase II (CKII) to phosphorylate the VDR on serine 51 and serine 208, respectively. A complex formed by the heterodimerization of the VDR with the RXR is translocated into the nucleus [31]. The translocated calcitriol–VDR–RXR complex regulates mRNA expression by binding to the VDRE and recruits transcriptional modulators to the promoter region of target genes [32]. Through direct protein–protein interaction, calcitriol binds to membrane-bound VDRs, or 1,25D-MARRS, in the non-genomic pathway. This causes acute changes in cell signaling pathways, including calcium and mitogen-activated protein kinase (MAPK) signaling [28,33] (Figure 2).

### 1.3. Is Vitamin D a Good Cancer Prevention Agent?

Vitamin D sensitizes cancer cells to the chemotherapeutic agent 5-fluorouracil (5-FU) by downregulating the expression of the antiapoptotic protein survivin and thymidylate synthase, a key enzyme involved in the biosynthesis of DNA. Additionally, in vitro studies have demonstrated that vitamin D promotes the differentiation of cells by increasing the expression of several cell adhesion components that are required for maintaining the epithelial phenotype along with proteins associated with the actin cytoskeleton and intermediate filaments. Treatment with vitamin D has been shown to significantly slow down tumor development and boost cancer cells’ susceptibility to chemotherapeutic drugs [34]. Epidemiological studies played a major role in the initial scientific discussion on vitamin D. The early studies were conducted in the United States, which provided evidence indicative of a north–south gradient in the risk of specific cancers. This finding has prompted the hypothesis of a protective influence of vitamin D on the risk of cancer at various sites. For instance, it has been demonstrated that vitamin D has a critical role in reducing the incidence of colorectal cancer [35]. Several observational studies have also reported a negative correlation between serum vitamin D levels and the risk of developing several cancers, including breast, colorectal, kidney, lung, and pancreatic [36]. Additionally, studies have reported a correlation between vitamin D deficiency and increased mortality due to several prevalent malignancies [37].

According to a meta-analysis, consuming enough vitamin D significantly lowers the risk and mortality of cancer [38]. Circulating vitamin D correlated well with patient survival, indicating that it may be a potential prognostic marker in advanced pancreatic cancer [39]. Yejin Kim et al. demonstrated that serum vitamin D could be used as a predictive marker for screening people at risk for CRC [40]. According to a case–control study of 102 participants, methylation levels of important CpG sites in VDRs, CYP24A1, and CYP2R1 are negatively correlated with the incidence of colorectal cancer (CRC) [41]. The CYP24A1 gene, which is located on chromosome 20, encodes for the 24-OHase. The enzyme 24-OHase is involved in the catabolism of 25(OH)D and 1,25(OH)2D and is responsible for the regulation of the concentration of these two vitamin D metabolites in the circulation. Studies have demonstrated that CYP24A1 is induced by 25(OH)D, 1,25(OH)2D, and FGF23. The CYP2R1 gene, which is located on chromosome 11, codes for an enzyme whose biochemical properties are consistent with a D 25-hydroxylase. The CYP2R1 gene transcript has characteristic sequence features associated with cytochromes P450 of the endoplasmic reticulum.

In a different study, Xuezhao Chen et al. used Mendelian randomization to examine the relationship between serum vitamin D levels and obesity as risk factors for basal cell carcinoma (BCC). This study found that a high BMI may increase the risk of BCC at the genetic level, but vitamin D has no effect on BCC [42].

Numerous epidemiological studies have also documented the association between vitamin D and different cancer types [43]. A cohort study of 182 children with neuroblastoma reported a lack of correlation between vitamin D deficiency and high-risk neuroblastoma [44]. Another study, which analyzed the function of vitamin D addition on telomere length, reported that administering monthly doses of vitamin D to older patients (age > 60) did not affect telomere length [45]. According to a cohort study involving 476 women with breast cancer, the patients with adequate vitamin D levels had smaller and lower-grade tumors than those with insufficient vitamin D (Table 1) [46]. Although these observational studies reported that serum vitamin D levels correlate with a lower incidence of cancers, they include various confounding factors like outdoor activity, adiposity, and overall nutrition status [47]. Hence, further evidence from RCTs is required to confirm these findings.

Several clinical trials testing the impact of vitamin D on cancer incidence, progression, and mortality have supported the benefits of vitamin D treatment and addressed some of its associated concerns [49]. Clinical trials conducted previously showed the antiproliferative effect of calcitriol in acute myeloid leukemia (AML) patients [50]. Several trials have successfully demonstrated a positive correlation between higher vitamin D levels in cancer patients and increased disease-free survival, lower recurrence risk, and reduced mortality rate [51]. One such study initiated in Norway, a country with minimal sun exposure, showed a better survival rate in cancer patients with higher vitamin D levels [52]. This study monitored vitamin D levels in a total of 658 patients suffering from breast (n = 251), colon (n = 52), lung (n = 210), and lymph (n = 145) carcinomas. The participants were divided into first (<2.5 mg/dL), second (2.5–3.4 mg/dL), third (3.4–4.5 mg/dL), and fourth (>4.5 mg/dL) quartiles based on circulating vitamin D. Death due to cancer was significantly lower (HR 0.36 95% CI 0.27, 0.51) in patients within the fourth quartile (>4.5 mg/dL) when compared to patients in the first and second quartiles. These results support a positive relationship between circulating 25-OHD and better survival in cancer patients [52]. Since the half-life of the active form of vitamin D (calcitriol) in systemic circulation is very short (15 h compared to 15 days for calcidiol), and the amount of circulating calcitriol is much lower (0.029 ng/mL to 0.083 ng/mL compared to 30 to 50 ng/mL of calcidiol). The levels of circulating vitamin D (calcidiol) rather than calcitriol are measured to determine the vitamin D status.

Robsahm, Trude Eid, et al. made an attempt to understand the effect of vitamin D in pre- and post-cancer diagnosis [53]. The study was conducted with 556 participants, which included breast cancer (n = 202), lung cancer (n = 193), lymphoma (n = 124), and colon cancer (n = 37) cases. Serum levels of 25-(OH)D were assessed before and after cancer incidence, and participants were divided into low (<2.5 mg/dL) and high (≥2.5 mg/dL) categories. The median 25-OHD levels were 3.5 and 3.4 mg/dL for pre- and post-diagnosis, respectively. Patients who had both samples at high (≥3.4 mg/dL) levels had a 59% lower mortality risk than the patients whose vitamin D levels were <2.5 mg/dL. Compared to individuals whose serum vitamin D levels were steady even after diagnosis, lower levels were linked to an increased risk of death [53].

A comprehensive meta-analysis comprising 23 case–control and 10 prospective cohort studies revealed a correlation between exposure and outcome, referring to circulating total 25(OH)D and CRC, respectively, in the overall population [54]. Whereas circulating 25(OH)D and 1,25(OH)2D were considered for both men and women in the case–control studies, only circulating 25(OH)D data were assessed for both men and women in prospective cohort studies [54]. Results of the case–control-based analysis showed a 39% lower risk of CRC in populations with serum 25(OH)D >33 ng/mL when compared to 12 ng/mL ((95% CI): 0.61 (0.52; 0.71); 11 studies) and a 20% lower risk of CRC in prospective studies (HR (95% CI): 0.80 (0.66; 0.97); 6 studies). However, these results were only observed in the female population and failed to show any significance in the male population [54]. Another separate analysis evaluated vitamin D levels in pooled data from 17 cohorts, which consisted of 5706 CRC patients and 7107 controls [55]. Based on circulating 25(OH)D levels, the study participants were divided into low-level (2.8– <3.4 mg/dL), deficient (1.6 mg/dL), and high-level (4.1–4.8 and 4.8– <5.5 mg/dL) groups. The deficient group had a 31% higher risk of CRC (RR = 1.31, 95% CI = 1.05 to 1.62). The higher level of vitamin D was linked to a lower risk of CRC. Interestingly, an inverse correlation of vitamin D level with CRC risk was observed only among the female participants but not in male participants [55].

#### Vitamin D Supplementation and Cancer

Rather than focusing on cancer incidence, several observational studies were carried out to examine any potential associations between vitamin D supplementation and cancer mortality [56]. For instance, a study conducted by Poole et al. demonstrated no significant changes in the mortality rate of women upon receiving 1 year of supplementation of vitamin D [57]. A retrospective study, which included a total of 308 breast cancer patients on Trastuzumab chemotherapy, showed that supplementation of <10,000 IU/week of vitamin D along with Trastuzumab improved the disease-free survival rate in HER2+ non-metastatic breast cancer patients when compared to those patients who did not receive the weekly vitamin D dosage [58]. In a separate study, the intake of vitamin D as a component of diet or as a supplement was studied in 434 head and neck cancer patients. Analysis of the data showed no significant effect on the mortality rate; however, lower disease recurrence was found in individuals with higher vitamin D levels [59].

A case–control study conducted in the US has evaluated the impact of vitamins and fatty acids on glioblastoma patients [60]. A total of 470 patients were considered, of which 77% of them were on alternative therapies. The addition of vitamin D has reduced the mortality rate of patients when compared to those who were not taking the supplement. However, this association was not observed after applying multivariate adjustments for factors like the KPS (Karnofsky Performance Scale) and tumor removal [60]. The use of de novo vitamin D among women between the ages of 50 and 80 years who were diagnosed with breast cancer was associated with a 20% decrease in the mortality rate [61]. Furthermore, a decrease of 49% in mortality rate was reported in patients who were put on vitamin D supplements soon after breast cancer diagnosis [61]. Another study measured the effect of vitamin D intake on the early inception of CRC [62]. A total of 111 women aged below 50 years were included in the study. The results of this study showed a significantly lowered risk of early onset of CRC with increased vitamin D intake (HR for ≥450 IU/day vs <300 IU/day, 0.49; 95% CI, 0.26–0.93; P for trend = 0.01). Furthermore, this association was evident when there was an increase in the dietary intake of vitamin D (HR per 400 IU/day increase, 0.34; 95% CI, 0.15–0.79) rather than vitamin D supplementation (HR per 400 IU/day increase, 0.77; 95% CI, 0.37–1.62), demonstrating that dietary intake is more beneficial compared to the supplements in reducing the risk of CRC onset [62]

Numerous RCTs have been carried out to evaluate the impact of vitamin D in the prevention of cancer; among them, only three RCTs published by Lappe et al. in 2007 [63], Bolland et al. in 2011 [64], and Lappe et al. in 2017 [65] have corroborated the findings of observational studies and documented a decrease in cancer risk in those using vitamin D supplements [66]. Nevertheless, the analysis in all three RCTs was limited to those who had not used calcium or vitamin D supplements before enrolling in the study [66]. Furthermore, these RCTs overlooked the fact that the duration of vitamin D (25(OH)D) in the blood, rather than the vitamin D dosage itself, influences the health outcome [67]. Taking these limitations into account, the biggest RCT, known as VITAL (NCT 01169259), examined the potential of a daily dose of 2000 IU vitamin D3 to reduce cancer risk over five years. The RCT included 25,871 participants. The findings of the VITAL trial revealed that vitamin D addition did not affect the incidence of invasive cancer in post-menopausal women [68]. In addition, the study findings showed that 25-hydroxyvitamin D levels were not associated with subsequent breast cancer risk [69]. A meta-analysis by Keum N., et al. 2019 reported that vitamin D supplementation (circulating levels of 25(OH)D around 3.0–7.4 mg/dL) significantly reduced total cancer mortality but failed to reduce total cancer incidence [70]. Virtanen J.K., et al., 2022 reported that vitamin D3 supplementation (1600 IU/day or 3200 IU/day) did not lower invasive cancer incidence among older adults, possibly due to sufficient vitamin D at baseline (4.1 mg/dL) in a majority of study participants [71]. Following these publications, arguments on vitamin D’s potential to prevent cancer have acquired even more traction. One of the potential reasons for not observing a positive correlation between vitamin D and a lower risk of cancer is that the vitamin D levels at baseline are too high in the participants to show the benefit of supplementation. In conclusion, further in-depth studies are warranted to find out whether taking supplements of vitamin D can help prolong the protection against cancers.

### 1.4. Darker Side of the Sunshine Vitamin

Although several epidemiological studies have shown a negative correlation between serum vitamin D levels and the incidence of cancer, little attention has been paid to the use of elevated vitamin D dosages in the treatment of cancer and its potential health consequences. According to research by Garland and colleagues, maintaining blood levels of vitamin D metabolites—which are essential to decrease the risk of breast and colon cancers—requires daily consumption of vitamin D in the range of 4000–8000 IU [72]. On the other hand, prolonged use of vitamin D supplements at a dosage exceeding 400–600 IU (i.e., RDA) raised blood vitamin D levels and resulted in hypervitaminosis. Although hypervitaminosis is more likely to occur when someone takes more than 10,000 IU of vitamin D per day, it is a possibility that excessive doses of vitamin D might be consumed either due to misuse of over-the-counter supplements or erroneous prescriptions [73]. Hypercalcemia, or an excess of calcium in the blood, is one of the main effects of hypervitaminosis. Excessive calcium in the blood can cause problems with the kidneys and bones, including kidney stones. Therefore, to prevent hypervitaminosis, individuals utilizing vitamin D supplements should measure their serum vitamin D. Innovative approaches are warranted to mitigate these side effects while maintaining the therapeutic level of vitamin D in serum. Using vitamin D derivatives, which do not raise blood calcium but encourage the death of cancer cells, is one possible strategy. The studies that tested the effects of vitamin D analogs on cancer incidence and treatment will be described in Section 2.

### 1.5. A Promising Potential of Vitamin D Supplement: Anticancer Effects

People who obtain enough vitamin D have lower cancer incidence and fewer cancer-related deaths. This has been shown in several epidemiological and observational studies [43]. Recent findings have provided support for this epidemiological observation by demonstrating greater death rates among those who received very little natural light in comparison to those who lived in higher-latitude locations, which are linked to higher levels of naturally synthesized vitamin D in the skin [74]. Studies have also shown that sunlight protects against the incidence of different cancers, including skin, prostate, colorectal, breast, and ovarian cancers [75]. Circulating vitamin D was shown to be correlated with a decreased prevalence of prostate and colorectal malignancies [76]. The first evidence of vitamin D growth inhibitory effects in tumor cells was shown by Colston and colleagues by demonstrating a pivotal role of VDR in malignant melanoma [77]. Subsequently, several other studies validated vitamin D’s antitumor properties [78,79,80,81,82,83,84,85,86,87,88]. Vitamin D and its analogs show their antitumor properties by (a) inhibiting cell proliferation, (b) inducing proapoptotic genes, (c) reducing angiogenesis, and (d) blocking the spread of cancerous cells [89]. Recent findings highlighted the growth-inhibitory properties of vitamin D in several cancerous cells and demonstrated the differentiation of leukemia cells (HL60) to macrophage lineage cells [90]. Several mechanisms and effectors, such as prostaglandins, COX-2, 15-PGDH, and EP2, were reported to be involved in the vitamin D-mediated inhibition of prostate cancer cell proliferation [91,92]. The combination of vitamin D with gemcitabine showed the altered gene expression of pancreatic stellate cells and minimized the tumor volume [93]. The proliferation of breast cancer cells was inhibited by vitamin D via selective downregulation of estrogen receptor alpha (ERα) in the malignant cells and blockade of aromatase expression in breast adipose tissue [94]. To mitigate the toxic effects, such as apathy, drowsiness, depression, psychosis, polydipsia, anorexia, constipation, peptic ulcers, hypercalciuria, polyuria, polydipsia, dehydration, nephrocalcinosis, and renal failure, which are associated with the administration of vitamin D, analogs of vitamin D are now being investigated. Vitamin D analogs are reported to be beneficial to health in preclinical and clinical studies [95]. For instance, several studies have demonstrated that vitamin D and analogs can trigger apoptosis in cancer cells [96]. Notably, the observed cancer cell death was induced through the (1) inhibition of angiogenesis and metastatic potential, (2) upregulation of proapoptotic gene expression, (3) activation of antitumoral immunity, and (4) chemotherapy sensitization [95]. Vitamin D-mediated inhibition of angiogenesis prevents the release of several growth- and survival-promoting factors and deprives cancer cells of oxygen and nutrients [97]. Mechanistically, the calcitriol–VDR complex induces conformational changes in the receptor structure, thereby promoting the binding of the VDR to the retinoic acid receptor (RXR). Interaction of VDRs with RXRs activates VDRE in the promoter region of various target genes, thereby controlling cell division, differentiation, proliferation, and metastasis [98].

### 1.6. Cancer Risk Reduction by Vitamin D Metabolite Calcitriol

Calcitriol activates genes that control stress response, DNA repair, immune responses, and various transcriptional factors responsible for the regulation of cellular processes [99]. These additional functions of calcitriol, which are not necessarily linked to the regulation of calcium homeostasis, represent a potential role of this vitamin D metabolite in the prevention and treatment of cancers [100]. The ability of calcitriol to resolve inflammation may be exploited for the inhibition of angiogenesis and cancer cell proliferation [101]. For instance, an enhanced antiproliferative effect was reported in ER+ breast cancer cells by calcitriol. Mechanistically, calcitriol downregulated aromatase transcription and blocked the synthesis of estrogen [102].

The activity of calcitriol is self-regulated by CYP24A1, which encodes an enzyme that catalyzes the degradation of both calcitriol and 25-hydroxycholecalciferol, thereby reducing the calcemic effects compared to vitamin D [103]. Notably, the presence of enzyme alpha 1-hydroxylase (encoded by CYP27B1) was reported in normal and malignant breast tissues [102]. The enzyme converts circulating prohormone 25(OH)D3 to the active hormone calcitriol in the breast tissue [102]. Therefore, the fortification of vitamin D in the diet helps to enhance the levels of substrate for CYP27B1, increase the production of calcitriol at the local tissue level, and inhibit the progression of breast cancer [102]. Accordingly, local production of 1, 25(OH)2D3 by alpha 1-hydroxylase was noted in breast cancer tissues [104]. Increased 1,25(OH)2D3 production due to dysregulated extra-renal alpha 1-hydroxylase expression was reported in B-cell lymphoma and breast, colon, and prostate cancers [105]. Increased expression (21-fold) of alpha 1-hydroxylase was also observed in less differentiated CRC tissues compared to normal adjacent tissue. However, highly differentiated CRC tissues did not show significant changes in alpha 1-hydroxylase levels, indicating that the enzyme expression might vary in different stages of cancer [106]. Interestingly, a case study showed that the abnormal synthesis of the enzyme depends on the paracrine activity of cancer-associated macrophages [107]. In conclusion, the local expression of alpha 1-hydroxylase may play an important role in tumorigenesis of several cancers, which warrants further investigations.

## 2. Vitamin D Analogs for Cancer Treatment

Even though vitamin D has been reported to have anticancer properties, some studies have documented negative consequences, such as hypercalcemia and an imbalance in the regulation of bone metabolism due to long-term usage of vitamin D. Hypercalcemia leads to clinical manifestations and symptoms of toxicity of long-term vitamin D usage, hence limiting its potential use as a supplemental chemotherapeutic drug for extended durations. Several vitamin D derivatives were produced to tackle various issues related to vitamin D-based therapy. The structure and anticancer effects of vitamin D and its analogs are shown in Table 2. Derivatives of vitamin D have relatively low calcemic efficacy and are being investigated in clinical studies for possible anticancer applications. When compared with the naturally occurring vitamin D, several of these analogs are more selective VDR ligands [108]. In the following sections, we describe vitamin D analogs and assess their prospective application in clinics.

### 2.1. EB-1089 (Seocalcitol)

Compared to vitamin D, EB-1089, a synthetic analog of vitamin D, has 50–100 times greater binding capacity to VDRs and a stronger ability to suppress cancer [111]. By initiating apoptosis in breast cancer cells (MCF-7) in vitro, EB-1089 caused tumor regression [111]. The anticancer effects of EB-1089 were mediated by Bcl-2/Bax activation [112]. Moreover, EB-1089 did not stimulate hyperglycemia but induced the expression of cell cycle inhibitors p-21 and p-27 [113,114]. EB-1089 also suppressed tumor growth via increased levels of microRNA (miR-498) while downregulating the expression of the hTERT gene in ovarian cancers [115]. EB-1089 regulated the activity and expression of several transcription factors in pancreatic cancers [116]. It has been observed that EB-1089 inhibits the growth of breast cancer cells both in vitro and in vivo [113,117]. The agent was shown to block the growth of H&N SCCs [114], HCCs [118], and ovarian [88,115], pancreatic [116], and non-small cell lung [115] cancers.

### 2.2. HY-11

The vitamin D-based agent demonstrated effective anti-leukemic activity [119]. Its ability to retard HL-60 cancer cell growth was attributed to G1 phase cell cycle arrest in a dose-dependent way. The antiproliferative effect of this compound was also mediated by the induction of apoptosis through the caspase 3 pathway and increased expression of transforming growth factor beta (TGFβ) [119]. However, the effects of this agent in vivo were not tested.

### 2.3. Tacalcitol

Coded as PRI-2191 or Tacalcitol, the 1α,24(R)(OH)2D3 analog of vitamin D was tested in CRCs in combination with standard chemotherapeutic drugs [109]. The anticancer activity of PRI-2191 in combination with 5-FU (5-fluorouracil) was observed in HT29 CRC cells. The agent upregulated CDKN1A expression through VDRs, leading to an ultimate decrease in the expression of thymidylate synthase [120]. Treatment with Tacalcitol sensitized HT29 cells to 5-FU therapy, supporting the high potential of this agent as a treatment for gastrointestinal malignancies [120].

### 2.4. Inecalcitol

Inecalcitol (TX-522) is an epi-analog of calcitriol. It is also known as Hyrigenics paris [121,122]. Inecalcitol inhibited the growth of squamous cell and prostate cancers by inducing caspase-3, 8/10-mediated apoptosis [122]. P-27 and P-21 are the proapoptotic effectors that were activated in prostate cancer by inecalcitol. In prostate cancer cells, inecalcitol reduced the expression of Pim-1 and Ets variant-1 [123]. Inecalcitol demonstrated advanced growth-inhibitory and VDR-binding properties compared to vitamin D in animal models. However, its effects on the level of blood calcium in humans remain unknown [123,124]. Administration of inecalcitol (80 µg/day up to 3 days) inhibited the growth of SCC [124]. Another study demonstrated that inecalcitol (1300 μg/kg up to 3 days) also inhibited the proliferation of SCC and did not induce hyperglycemia [123]. When administered in combination with docetaxel, inecalcitol was well tolerated in individuals with prostate cancer; nevertheless, hypercalcemia caused dose-limiting toxicity [123]. In a phase-2 clinical study, individuals with myeloid leukemia are being treated with inecalcitol in addition to decitabine (NCT02802267).

### 2.5. TX527

TX527 was shown to activate the VDR/RXR complex at lower doses than vitamin D [109]. The antiproliferative effect of this agent was tested in Kaposi’s Sarcoma-associated herpesvirus GPCR-transformed endothelial cells (SVEC-vGPCR) [125]. TX527 effects in SVEC-vGPCR cells were marked by the inhibition of NF-κB expression (similar to that of vitamin D) and increased expression of IκBα. Activated IκBα decreased the localization of NF-κB in the nucleus and, thus, prevented its gene-activating effects. After receiving TX527, there was a decrease in the synthesis of inflammatory cytokines (IL-6, CCL2, and CCL20). Therefore, the agent demonstrated effective antiproliferative and anti-inflammatory properties [125], although further clinical testing is warranted.

### 2.6. Paricalcitol

The anticancer activities of Paricalcitol were tested in different cancer cells [126,127,128]. The agent stimulated leukemia cell HL-60 differentiation (maturation), which was marked by the elevated expression of the CD11b marker and decreased colony formation (prolonged effect after 10-day treatment) [126]. Incubation of HL-60 cells with Paricalcitol (72 h) increased apoptosis and cell cycle arrest at the G0/G1 phase [126]. Paricalcitol inhibited the growth of gastric cancer cells (AGS, MKN45, and SNU719) by inducing apoptosis, which was marked by activation of caspase 3 and decreased Bcl-2 protein expression [127]. An in vivo CRC growth study tested the combined treatment effect of 5-FU and Paricalcitol [128]. The data showed decreased tumor growth in rats that received the combination therapy compared to the monotherapy-treated animals. The combined drug treatment triggered Wnt/β-catenin, NF-kB, and COX-2 signaling cascades, which are involved in the initiation and progression of CRCs [128].

### 2.7. Doxercalciferol

Doxercalciferol (1α,25-(OH)2D2) is a synthetic vitamin D2 analog, which can be metabolized to form an active form of vitamin D2 in vivo. The agent was tested in combination with an arsenic compound KML001 in acute lymphoid leukemia models. The combined application demonstrated a synergistic effect and increased the number of cells in the late apoptotic phase [129]. However, the agent was not tested in other solid tumor models.

### 2.8. Maxacalcitol

Maxacalcitol is a non-calcemic analog of vitamin D3. The anticancer effect of this agent was studied in pancreatic cancer cells. In vitro, the analysis showed G1 phase cell cycle arrest, while in vivo, the BxPC-3 cell xenograft model showed retarded tumor growth without hypercalcemia [130]. Further analysis is warranted.

### 2.9. Calcipotriol

It was discovered that calcipotriol, also known as PRI-2201, inhibits the development of tumors in animal models of pancreatic cancer [93,131]. Calcipotriol administration increased stromal remodeling and promoted the uptake of the anticancer drug gemcitabine. Accordingly, tumor regression was detected in vivo in animals treated with a combination of calcipotriol. Moreover, calcipotriol alone can also inhibit tumor growth via decreased Wnt/catenin signaling [131]. The agent was not tested against other cancers.

### 2.10. BGP-13

BGP-13 is a synthetic calcipotriene-based analog of vitamin D3. The anticancer activity of BGP-13 was tested in pancreatic (LNCaP), breast (MCF-7), and colon cancer (HT29) cell lines [132]. The agent can induce apoptosis and cell cycle arrest at the G0/G1 phase. Apoptosis was activated by BGP-13 through the caspase-3 pathway. The agent stimulated the expression of VDRs in MCF-7 and LNCaP cells. HT29 xenografts in nude mice were also sensitive to the growth-inhibiting effects of BGP-13 [132].

### 2.11. PRI-2205

PRI-2205 was tested for its anticancer activities in vitro and in vivo [133]. Combined treatment with cisplatin showed a promising cytostatic potential of this agent. In combination with tamoxifen, PRI-2205 induced G2/M phase cell cycle arrest in MCF-7 breast cancer cells [134]. Another study reported a strong reduction in mRNA levels of estrogen receptors (ERs) in MCF-7 cells treated with a combination of PRI-2205 and anastrozole (an aromatase inhibitor) [133]. The combined treatment showed the reduced expression of other genes involved in the ER signaling, including estrogen-related receptor alpha (*ESRRα),* estrogen-related receptor gamma (*ESRRγ),* and Gonadotropin-releasing hormone *(GnRH1)* [133]. Therefore, the agent provoked substantial interest as a potential regulator of gene expression in cancer cells.

### 2.12. PRI-1906

PRI-1906 stimulated an increased mRNA expression of CYP24A1 in patient-derived ovarian cancer cells. The agent increased the level of nuclear VDRs, which led to decreased cell viability [135]. Further testing of this agent in different cancer models is warranted.

### 2.13. BXL-01-0126

BXL-01-0126 belongs to the C20 Gemini class of vitamin D3 analogs. It exhibits better anticancer activity than vitamin D3, although similar hypercalcemic properties were registered [136]. In AML cells, the agent increased cAMP levels in a dose-dependent manner. Similar results have been observed in vivo using a xenograft model. The agent’s ability to regulate cAMP levels in AML patients may protect them from microbial infections during chemotherapy [136].

### 2.14. BXL0124

BXL0124 has demonstrated anticancer activity in different cancer models in vivo, including breast cancer xenograft models [137,138,139]. BX1024 blocked the local invasion of tumor cells in the preclinical testing [140]. However, the mechanism of BXL0124’s anti-metastatic effects remains unclear and warrants additional investigations.

### 2.15. Gemini0097

Gemini0097 belongs to the family of the C20 class of vitamin D3 analogs. It is a less toxic and more efficient anticancer substance than vitamin D3 [141]. The agent was tested in vivo using an ER-positive breast cancer model with chemically induced tumors (N-methyl-N-nitrosourea (NMU)-induced cancer). Gemini0097 was the most potent agent among the tested substances and caused up to 60% growth inhibition of NMU-induced tumors. The agent also increased the expression of p21 (CDK inhibitor) and insulin-like growth factor binding protein 3 (IGFBP3) in ER+ and ER^−^ tumors [142].

### 2.16. MART-10

MART-10 is a potent anticancer agent without any serious side effects. The agent was tested in MDA-MB-231 (triple-negative breast cancer (TNBC) cell line) [143] and decreased the migratory and invasive potential of these cells. These effects were mediated by the upregulated E-cadherin and downregulated N-cadherin proteins. MMP9 downregulation was also observed upon treatment with MART-10. These results support the promising potential of this substance for the treatment of TNBC [143]. However, further investigation is required to ensure its effectiveness in vivo.

### 2.17. 1,25-Dihydroxyvitamin D_3_-3-Bromoacetate

The anticancer effect of 1,25(OH)_2_D_3_-3-BE was evaluated in the human renal tumor (A498 and Caki1) xenograft model. The agent was a more effective inhibitor of tumor growth compared to vitamin D3 in that study [144]. It also effectively inhibited cell cycle progression via enhanced caspase activity, decreased expression of cyclin A, and the inhibition of Akt phosphorylation [144].

### 2.18. Ro26-2198

Ro26-2198 delayed the onset of colitis in an azoxymethane (AOM)-induced CRC model in mice. The anticancer property of this agent in CRC was attributed to the decreased expression of c-Myc, COX2, and pERK (oncogenes) in vivo. In vitro models confirmed the observed mechanism of Ro26-2198 effects. Moreover, the agent decreased the expression of IL-1B, suggesting a possible role in the regulation of inflammation [145].

### 2.19. EM1

EM1, the more stable analog of vitamin D, demonstrated antiproliferative properties in vitro in a variety of cancer cell types [146,147]. The administration of EM1 demonstrated antitumor activity in animal models with various cancer xenografts, including breast carcinomas. The administration of EM1 inhibited metastasis by E-cadherin expression [148].

## 3. Molecular Mechanisms of Tumor Growth Inhibition by Vitamin D

### 3.1. Molecular Mechanisms of Vitamin D-Induced Apoptosis

Morphologic changes associated with apoptosis include cell shrinkage, blebbing of the plasma membrane, release of mitochondrial cytochrome c, fragmentation of the cellular DNA into multiples of 180 bp, and, finally, the breaking of the cell into small apoptotic bodies that will be cleared by nearby cells through phagocytosis. The activation of caspases (cysteine aspartate-specific proteases) is known to cause these morphologic alterations [149]. Apoptosis was indicated as an important mechanism of anticancer effects induced by vitamin D and its analogs [150]. However, there are several conflicting results on vitamin D-induced apoptosis. For instance, one study demonstrated that vitamin D induces apoptosis via a caspase-dependent mechanism [151], while others reported activation of caspase-independent cascades [152,153]. In MCF-7 cells, vitamin D induces apoptosis via a caspase-independent mechanism by activating mitochondrial malfunction, translocating Bax, and generating reactive oxygen species (ROS) [153]. But a separate study reported that vitamin D3 induces apoptosis in breast cancer cell lines (MCF-7 and in MDA-MB-231) by triggering Caspase-3/7 [154]. The vitamin D analog EB-1089 induced apoptosis in MCF-7 cells via elevating intracellular calcium levels, which activated μ-calpain [155]. The activation of autophagy (as evidenced by enhanced beclin-1 expression) by vitamin D was also reported in certain studies, which was further linked to the upregulation of apoptosis [156]. By increasing the proapoptotic BAK gene and downregulating the antiapoptotic proteins Bcl-2 and IAP, vitamin D induces apoptosis in CRC cells [157]. Furthermore, vitamin D can induce apoptosis in prostate cancer cell lines (LNCaP and ALVA-31) by activating mitochondria-related apoptotic pathways [151]. A downregulation of telomerase (hTERT), an activation of caspase-dependent MEK cleavage, and the overexpression of p53 are the other key mechanisms involved in vitamin D-induced apoptosis [83,158].

### 3.2. Antiproliferative Mechanisms of Vitamin D

Deregulated proliferation is one of the hallmarks of cancer cells [159]. Cancer cells not only generate specific growth factor production in stromal cells but also alter growth factor signaling pathways [159]. Notably, vitamin D-induced effects were shown to trigger numerous signaling mechanisms that stimulate growth inhibition and apoptosis. Previous research has demonstrated that vitamin D activates p-21waf1/cip1 and induces G0/G1 cell cycle arrest, which results in antitumor actions in breast and prostate cancer cells [160]. VDRs and VDRE directly target P-21waf1/cip1 [161,162]. By increasing the expression of p21waf/cip1 and p-27kip1, as well as by lowering the levels of cyclins and cyclin-dependent kinases (CDKs), vitamin D treatment inhibits pancreatic cancer cells [163]. Another study discovered that vitamin D administration causes squamous cell carcinomas to produce more p-27kip1 and p21wf/cip1, which in turn causes G0/G1 phase cell cycle arrest [164].

Vitamin D treatment inhibited the growth of ovarian cancer cells by inducing G2/M phase arrest, which is triggered by transcriptional activity that is independent of p53 [165]. Furthermore, vitamin D administration could cause cell cycle arrest by lowering ERK1/2 expression and activity [166]. The vitamin D analog EB-1089 also showed growth-inhibitory potential. The effect was mediated by the induction of PTEN, which inhibited the Akt pathway in thyroid carcinoma cells [167]. In animal models of thyroid carcinogenesis, vitamin D inhibited PIK3/Akt-associated cell proliferation. [168]. Vitamin D and its analog also demonstrated antitumor activity in kidney cancer cells, which was mediated by the inhibition of Akt and its target gene caspase-9 [144]. Furthermore, vitamin D treatment arrested the growth of endothelial cells by inhibition of NF-kB activation, a mechanism independent of PI3K/Akt and MAPK pathways [169]. Growth-inhibitory effects of vitamin D treatment in MCF-7 cells were mediated by Src-tyrosine kinase and tyrosine phosphatase activation and the phosphorylation of ERK1/2 [170]. Another study reported that vitamin D inhibits the proliferation of breast cancer cells via activating IGFBP3, a target of the VDR and a functional activator of VDRE [171]. Vitamin D treatment elevates TGF-β production, which is another potent regulator of proliferation and apoptosis in breast cancer cells [172]. Vitamin D treatment retarded the proliferation of CRC cell line SW-480-ADH via upregulation of microRNA-22 (miR-22) expression [173]. Increased levels of let-7a-2 (anticancer miR) were reported in A-549 cells [174]. The effects were associated with the activation of VDRE [174]. However, many of the reported effects were observed in vitro and require further clinical confirmation.

### 3.3. Vitamin D Inhibits Key Events in the Metastatic Spread of Cancer Cells

Metastasis is a complex process by which cancer cells spread from the original tumor to form a new tumor in other parts of the body. Cell adhesion, invasion, migration, and re-establishment are the key steps involved in metastasis [175]. The spread of cancer to distant organs is responsible for almost 90% of cancer-related fatalities [176]. Shreds of evidence reported that vitamin D and its analogs modulate several crucial steps of metastasis. For instance, it has been demonstrated that EB-1089 activates fibronectin in thyroid cancer cells in a PTEN-dependent manner, thereby restoring cell adhesion [177]. In squamous cell carcinoma (SCC), vitamin D was shown to inhibit tumor cell motility via modulating E-cadherin expression [178]. Vitamin D inhibited the invasion and metastasis of cancer cells by downregulating the secretion of MMP-2 and MMP-9 [178]. The invasion of prostate cancer was reduced by vitamin D through the reduction in MMP-9 and cathepsins secretion, while MMP-1 (a tissue inhibitor) was activated [179].

The metastatic potential of prostate cancer cells was reduced by vitamin D, which decreased the adhesion and rolling activity of cells in the vasculature [180,181]. In Lewis lung carcinoma cell lines (LLC and LLC-LN7), vitamin D treatment blocked the invasion and migration by decreasing the production of GM-CSF, which was associated with reduced activity of PKA [182,183]. Other mechanisms of inhibition were associated with the downregulation of Dickkopf 4, a protein-coding gene, which inhibits the Wnt/β-catenin signaling and MAPK pathways [184]. By inhibiting SNAI1, Slug, and vimentin as well as the EMT, the vitamin D analog MART-10 reduced the migration of pancreatic cells BXPC-3 and PANC [185]. Interestingly, vitamin D facilitated cell motility in normal cells by the specific activation of PI3K [186].

Vitamin D treatment also inhibited the migration and invasion of cancer cells in animal models [187]. In a melanoma B-16 mouse model, vitamin D (daily at a dose of 0.5 µg/kg for 28 days) decreased both experimental and spontaneous pulmonary metastases [187]. Vitamin D analog 22-oxa decreased the cancer colony formation in the lungs by negatively regulating fibroblast-induced angiogenesis [188]. In mice models of breast cancer, another vitamin D analog, EB-1089, had inhibitory effects on bone metastasis [189]. Larger studies are warranted to confirm the observed anti-metastatic effects of vitamin D analogs.

### 3.4. Vitamin D Induces Cancer Cell Differentiation

Cancer cells have lost the ability to differentiate. This characteristic allows the malignant cells to grow faster and spread quicker compared to normal/differentiated cells [190]. Acute myeloid leukemia (HL-60) cells treated with vitamin D were seen to exhibit a mature monocyte phenotype, suggesting that vitamin D can induce cell differentiation [191]. The ability to differentiate upon exposure to vitamin D was marked by cell cycle arrest and elevated p-21waf1/cip-1 and p-27kip-1 [191]. In human monocytic leukemia cells, vitamin D promoted differentiation through the inhibition of p-38 MAP-kinase or activation of PI3K [192,193]. However, another study reported the involvement of JNK and MAPK signaling cascades in vitamin D-mediated differentiation of HL-60 and U-937 cells [194]. Vitamin D activated ERK1/2 in HL-60 cells, which in turn upregulated the expression of C/EBPβ and c-JUN [195]. Enhanced nuclear localization of C/EBPβ-2 and C/EBPβ-3 promoted the expression of Ras-1, a well-known differentiation trigger [196].

Earlier reports stated that vitamin D treatment promotes epithelial carcinoma differentiation through the activation of alkaline phosphatase and AP-1 protein [197]. Vitamin D inhibited cell migration by increasing the expression of E-cadherin and decreased the levels of vimentin, SNAI1, and ZEB1 [198]. For example, vitamin D suppressed the expression of mesenchymal markers (N-cadherin, p-cadherin, and integrins α-6 and β-4) but increased the expression of claudin-7 and β5 FAK in MDA-MB-453 breast cancer cells [199]. Vitamin D analog 1α-(OH)D5 showed the differentiation of breast cancer cell line T47D by increasing lipid production and casein expression [200]. Vitamin D’s ability to stimulate differentiation in MCF-7 and T47D mammary cancer cell lines was mediated not only by cell cycle arrest but also by reducing the ability to produce anchorage-independent colonies [200,201]. Vitamin D activated the pro-differentiation-related gene ICB-1, which subsequently increased the expression of E-cadherin in MDA-MB-231 breast cancer cells [202]. Vitamin D was also found to upregulate the expression of androgen receptor (AR) and PSA proteins, which can regulate the differentiation process [203]. However, the role of vitamin D analogs in the regulation of cell differentiation remains unclear and warrants further investigation in vivo and in clinical studies.

### 3.5. Vitamin D Inhibits Angiogenesis and Constrains Tumor Growth

According to earlier observations, solid tumors are incapable of expanding their diameter beyond 2.0 mm if they are unable to induce the growth of new blood vessels [204]. For malignant cells to survive and proliferate, an abundant supply of nutrients and oxygen is necessary, for which tumor cells promote the formation of blood vessels (angiogenesis). The process is crucial for tumor progression, metastasis, and the development of drug resistance [205,206,207,208]. Vitamin D was shown to influence angiogenesis via multiple mechanisms. For instance, TX-527 (vitamin D analog) downregulated the expression of G protein-coupled receptors (GPCRs), which led to the inhibition of murine endothelial angiogenesis [209]. The agent significantly inhibited vascular GPCR-related tumor progression in vivo [209]. Vitamin D treatment suppressed the expression and signaling of vascular endothelial growth factor (VEGF), thus triggering apoptosis and inhibition of vascular cell elongation [210]. In SCC models, the proliferation of tumor-derived endothelium cells was also inhibited by other vitamin D analogs, including EB-1089, -6760, and -7553 [206].

Notably, apoptosis and cell cycle arrest in the G0/G1 phase were detected in tumor-derived endothelial cells (TDECs) exposed to vitamin D, while no significant death-promoting effects were observed in specific matrigel-derived endothelial cells [208]. Vitamin D treatment resulted in the increased expression of p-27kip-1, while Akt and ERK1/2 activities were downregulated [166]. Interestingly, no inhibition of vascular cell growth was observed in the experiments with embryonic chorioallantoic membranes (CAMs) exposed to vitamin D. However, numerous reports on cell-based and animal models have supported the anti-angiogenic effects of vitamin D and its analogs [211]. For instance, vitamin D blocked vascular growth in xenografted breast cancer models [210]. Topical administration of vitamin D also demonstrated anti-angiogenic effects in a mouse model with suture-induced corneal inflammation [212]. Treatment with vitamin D and its analog 22-oxa-1,25D3 decreased MMP-2, MMP-9, and VEGF levels [213]. In prostate cancer tumor models, vitamin D treatment was reported to downregulate MMP-9 and IL-8 levels, as well as inhibit the migration and tube formation in human umbilical vein endothelial cells (HUVECs) [214] (Figure 3).

## 4. VDR Gene Polymorphism in Cancers

Numerous biological processes, such as immunological responses, bone metabolism, specific cell proliferation, and differentiation in healthy tissues, are regulated by vitamin D. It has been demonstrated that during carcinogenesis, vitamin D regulates the genes involved in inhibiting the development, migration, adhesion, and angiogenesis of cancer cells [77,187,215,216,217,218,219,220,221]. Considering that VDRs mediate vitamin D biological activities, variations in VDRs can modulate the function of vitamin D [222,223]. Notably, the VDR-coding gene is known to contain about 200 single nucleotide polymorphs (SNPs) [224]. Polymorphic VDRs may have an impact on a person’s vitamin D levels. The alteration in VDR signaling is caused by SNPs in the VDR gene [225,226]. *Fok1* (rs10735810), *Bsm1* (rs1544410) [227,228], *Apa1* (rs7975232), *Taq1* (rs731236), and *Cdx2* (rs11568820) [229] genes are among the most extensively studied VDR SNPs associated with cancer.

Although further research is necessary to fully comprehend the role of VDR polymorphisms in different cancers, genetic VDR variations are critical markers used for the selection of better treatment strategies. For instance, the results of a meta-analysis indicated that VDR polymorphisms could be associated with a greater risk of keratinocyte cancers [230]. The SNP in the Cdx2 (Caudal-type homeobox protein 2, which is an intestine-specific transcription factor with a polymorphic binding site in the VDR gene) has been associated with an overall increased risk of cancer [229]. The Taq1-targeting VDR polymorphism was linked to an increased risk of colorectal cancer [229]. However, no correlation was observed between prostate cancer and polymorphisms in the Apa1 and Cdx2 VDR genes [231]. However, to support this conclusion, additional evidence is needed [231].

Another study suggested that the Fok1 VDR polymorphism might be a viable target to predict the risk of prostate cancer [232]. According to an updated investigation, there is a possibility that the Fok1 VDR polymorphism raises the risk of prostate cancer in Caucasians. To verify these results, more population-based studies are warranted to confirm these data [233]. On the other hand, a research investigation of a subset of Caucasians revealed no correlation between the risk of breast cancer and the allele contrast for the Apa1, Fok1, Tag1, and Bsm1 VDR gene polymorphisms [234]. However, the Fok1 VDR polymorphism was associated with ovarian and breast cancers in another study [235]. Liu et al. (2017) showed a substantial correlation between Asian and African American men’s prostate cancer risk and the rs731236 VDR [236]. The development of breast tumors may be influenced by polymorphisms in the Apa1, Bsm1, Fok1, and Poly(A) VDR genes, according to a different systematic review and meta-analysis [237]. Further studies are warranted to confirm this conclusion.

Another study reported an absence of correlation between TaqI polymorphisms and susceptibility to CRC [238]. Among the Asian population (particularly Japanese people), the Taq1 polymorphism was linked to an increased incidence of prostate cancer [239]. Prostate cancer that progressed to an advanced stage was more common in those individuals carrying the T allele or TT genotype. Consequently, VDR-linked Taq1 polymorphisms might be regarded as a possible diagnostic biomarker for the susceptibility to prostate cancer [239]. Despite the need for more research, a recent study found that VDR polymorphism is likely to increase the risk of lung cancer [240].

The Fok1 polymorphism was correlated with the type and severity of CRCs [241]. It has been shown that *Fok*1 and *Bsm1* are risk factors for CRC [242]. Moreover, a meta-analysis study indicated that the polymorphism of Fok1 may be linked to ovarian cancer in Caucasian populations [243]. However, the hypothesis was not confirmed in another study, which reported that the Bsm1 polymorphism is likely the best risk indicator of ovarian cancer in Caucasian patients [244]. The risk of breast and ovarian cancers was associated with *Fok1* but not with *Apa1* (rs7975232), *Cdx2* (rs11568820), and *Taq1* (rs731236) VDR polymorphisms, as indicated by a meta-analysis of published data [245]. Interestingly, the Bsm1 polymorphism was linked with a lower incidence of these malignancies [245]. Asian populations are more susceptible to renal cell cancer (RCC) due to the VDR gene polymorphisms Apa1 and Fok1 (FF genotype) [246]. According to a meta-analysis, polymorphisms in Bsm1, Cdx2, and Taq1 increase the risk of developing lung cancer [247]. Accordingly, the VDR Taq1 polymorphism was linked to an increased risk of malignancies related to tobacco usage and smoking [248].

A study suggested that Apa1, Bsm1, and Fok1 polymorphisms might influence the development of melanoma [249]. However, the presence of specific *Cdx2* and *Bsm1* variants was associated with a lower risk of lung cancer, while *Taq1* was associated with an increased risk of this cancer. The presence of AA genotypes of *Bsm1 and Apa1* variants was considered to be protective against lung cancers, whereas *Taq1* and *Fok1* polymorphisms were estimated as risk factors in Asian populations for this type of tumor [250].

VDR polymorphism (Apa1, Cdx2, and Taq1) is associated with an increased risk of developing various cancer types, including CRC [229], BCC, SCC, HCC, head and neck cancers, kidney cancers, and thyroid cancer. VDR gene RFLPs were also associated with an increased risk of 19 different types of cancers [251]. Therefore, strong associations were shown for breast (Apa1, Bsm1, Cdx2, Fok1, and Taq1), colorectal (Apa1, Bsm1, Fok1, and Taq1), prostate (Apa1, Bsm1, Cdx2, Fok1, and Taq1), and skin (Bsm1, Fok1, and Taq1) cancers [251]. Future research should focus on combining VDR-specific genetic polymorphisms with the measurement of vitamin D levels, with ethnicity as a stratum for the study.

Consequently, a comprehensive meta-analysis assessed the possibility of using VDR polymorphisms in diagnostics. After analyzing the data collected from 192 independent studies (98,209 cancer-free controls and 78,628 cases), [252] it was shown that Fok1, Bsm1, Cdx2, Apa1, and Taq1 are good markers for CRC, lung, ovarian, skin, multiple myeloma, and brain tumors [252]. When compared to other ethnic groups, Caucasians showed the strongest correlation. To determine the important correlations and causal relationships between VDR polymorphisms and particular cancer types, more research with larger cohorts is warranted. Studies depicting the role of VDR polymorphisms in various cancers are shown in Table 3.

## 5. Conclusions and Future Directions

Vitamin D has been recognized as a powerful anticancer agent because of its numerous inhibitory actions on tumor cells. Although several studies have demonstrated the anticancer effects of vitamin D and its analogs in lab-based and animal models, further studies are still required. Several controversies and concerns must be resolved before vitamin D can be considered for cancer treatment. The question “Is vitamin D a good cancer prevention or treatment agent?” is yet to be answered. Consequently, apart from deciphering the processes via which vitamin D and its analogs impede the proliferation of cancer cells, future studies should focus on why vitamin D by itself and in conjunction with other vitamins, such as vitamin E, has failed in reducing cancer incidence. The development of optimized versions of vitamin D and vitamin D carriers with cancer-cell-targeting capabilities should be the main emphasis of future research.

## Figures and Tables

**Figure 1 cancers-16-03211-f001:**
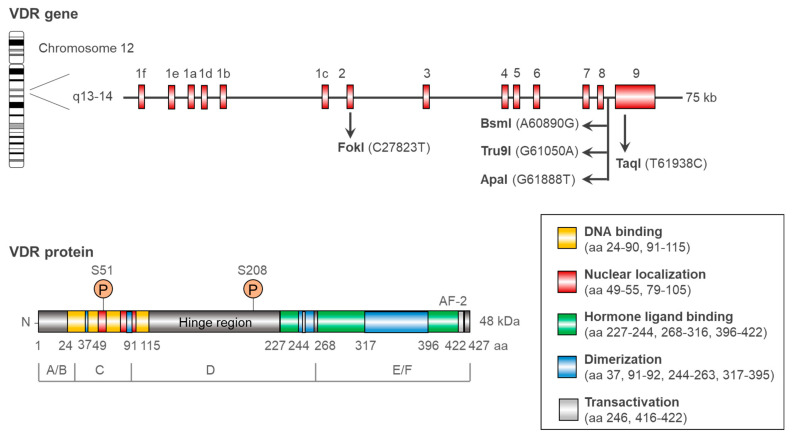
Schematic depiction of the VDR gene and protein. The q13–q14 arm of chromosome 12 contains the 48 kDa VDR protein-encoding gene. The VDR gene consists of a promoter and regulatory regions 1a through 1f. The VDR gene has eight exons (2–9) that together encode the six domains (A–F) of the complete VDR protein. The VDR protein has many domains, including dimerization, transactivation, hormone ligand binding, nuclear localization, and DNA-binding domains.

**Figure 2 cancers-16-03211-f002:**
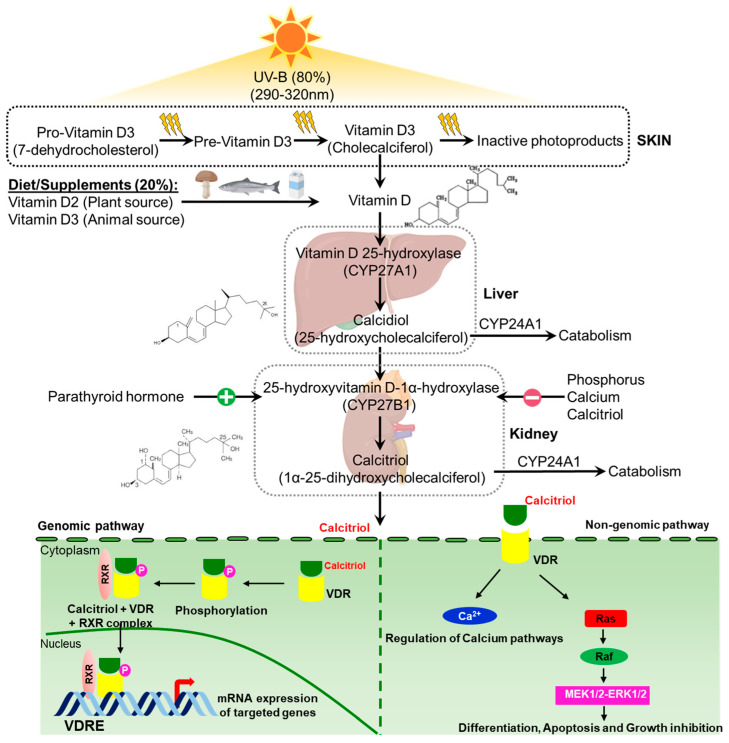
Vitamin D metabolism. In the liver, the dietary forms of vitamin D (vitamin D2 and vitamin D3) are metabolized into calcidiol by vitamin D 25-hydroxylase. Calcidiol is further metabolized by 25-hydroxyvitamin D3-1α-hydroxylase into calcitriol in the kidney. Calcitriol (active form) regulates the downstream targets by genomic and non-genomic pathways. In the genomic pathway, calcitriol binds to cytosolic VDRs, which promotes the phosphorylation of VDRs and heterodimerization with RXR. The complex then binds to VDREs in the nucleus and regulates the mRNA expression of the target genes. In the non-genomic pathway, calcitriol specifically binds to membrane-bound VDRs and regulates calcium and MAPK signaling cascades.

**Figure 3 cancers-16-03211-f003:**
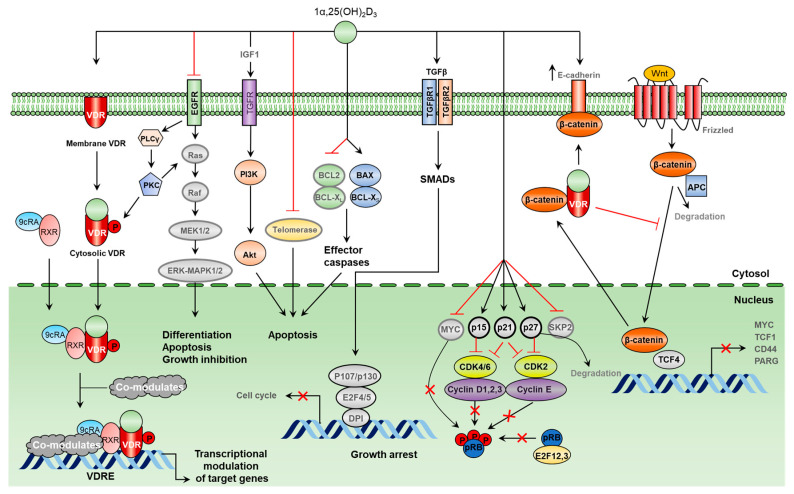
Key signaling pathways involved in the vitamin D-induced effects in different cancers. Antiproliferative effects of vitamin D are mediated by VDR-dependent and VDR-independent mechanisms. Vitamin D was shown to signal via various cell-surface receptors and impact the activation of telomerase, which is involved in the regulation of cancer cell proliferation and apoptosis. Vitamin D/VDR/VDRE/RXR complexes can trigger the expression of several cell cycle regulators and apoptosis modulators. Vitamin D was also shown to impact β-catenin-Tcf4 complex formation, which subsequently regulates cell cycle/apoptosis, Myelocytomatosis (Myc), T-cell factor (Tcf1), cluster of differentiation 44 (CD44), and Poly(ADP-ribose) glycohydrolase (PARG).

**Table 1 cancers-16-03211-t001:** Recent observational studies with vitamin D analogs.

Sl. No.	Study Design	Year	Sample Size	Conclusion	References
1	Case–Control Study	2023	293 (143 gastric cancer patients and 150 controls)	VDR Fok1 polymorphism is significantly associated with GC risk in the Kashmiri population	[48]
2	Ancillary Study	2023	1519 participants (vitamin D: n = 744; placebo: n = 775)	Vitamin D supplementation in older adults with vitamin D deficiency has no effect on the telomere length	[45]
3	Case–Control Study	2023	204 (cases—102; controls—102)	Methylation levels of significant CpG sites in VDRs, CYP24A1, and CYP2R1 are inversely associated with CRC risk	[41]
4	Cohort Study	2023	236,382 participants	Study showed the beneficial association of Serum 25(OH)D with risk of developing CRC.	[40]
5	Prospective Cohort Study	2023	476 women with incident stage I–III breast cancer (BC)	Women with sufficient vitamin D had smaller and lower-grade tumors compared to the women with insufficient vitamin D	[46]

**Table 2 cancers-16-03211-t002:** The anticancer effects of vitamin D analogs.

S. No.	Name	Structure *	Type ofCancer	AnimalType	InductionMethod	ResultsObtained
1	EB-1089(Seocalcitol)	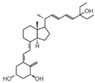	Breast cancer HCC	Mice (I)Mice (I)	SubcutaneouslySubcutaneously	Tumor growth inhibition
2	HY-11	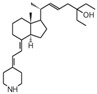	Mice were inoculated with leukemia cells	Mice (I)	Intraperitoneally	
3	Tacalcitol (PRI-2191)	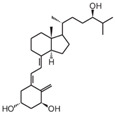	Colorectal cancer	Mice (I)	Subcutaneously	Tumor growth inhibition
4	Inecalcitol	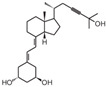	Squamous cell carcinoma	Mice (I)	Subcutaneously	Inhibition of tumor growth, increased apoptosis, and decreased proliferation
5	TX527	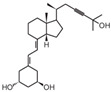	Kaposi’s sarcoma	Mice (I)	Subcutaneously	Tumor growth Inhibition
6	Paricalcitol	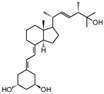	Metastatic breastcancer	Mice (I)	Subcutaneously	Tumor inhibition was accompanied by in vivoupregulation of p21 and p27 expression
7	Doxercalciferol	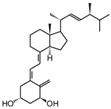	Neuroblastoma	Mice (I)	Flanks	Tumor growth inhibition
8	Maxacalcitol	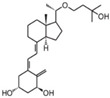	Cholangial carcinoma	Mice (I)	Subcutaneously	Inhibition of tumor growth and inhibition of proliferation
9	Calcipotriol	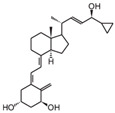	Non-melanoma skin cancer	Mice (I)	Subcutaneously	Tumor growth inhibition
10	BGP-13	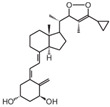	Colorectal cancer (CRC)	Mice (I)	Subcutaneously	Inhibition of growth of HT-29 tumors in mice
11	PRI-2205	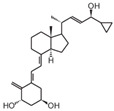	Breast cancer	Mice (I)	Subcutaneously	Lowering the expression of estrogen receptors and aromatase activity
12	PRI-1906	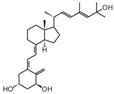	Breast cancer	Mice (I)	Orthotopically	Tumor growth andmetastases inhibition
13	BXL-01-0126	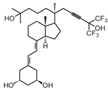	Acute myeloid leukemia	Mice (I)	Intrahepatic (IH) or facial (FV) vein	Activation of apoptosis
14	BXL0124	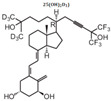	Breast cancer	Mice (I)	Mammary fat pads	Proliferation, angiogenesis, invasion, and metastasis
15	Gemini0097	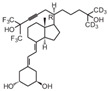	Breast cancer	Mice (I)	Mammary fat pads	Suppressed tumor growth and inhibition of tumor burden
16	MART-10	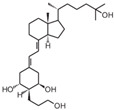	Pancreatic cancer	Mice (I)	Subcutaneously	Inhibition of tumor growth
17	(1,25(OH)2D3-3-BE)	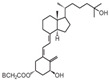	Kidney cancer	Mice (I)	Subcutaneously to the flanks	Inhibition of tumor growth and increase in apoptosis
18	Ro26-2198	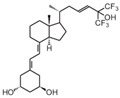	Colorectal cancer (CRC)	Mice (C)	Administration of Dextran sulfate sodium (DSS)	Inhibition of dysplasiaprogression and inhibition ofproliferation andpro-inflammatory signals
19	EM1	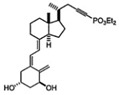	Breast cancer	Mice (I)	Subcutaneously	Reduced the formation of metastasis

* The structure formulas have been sourced from PubChem and Tocris Bioscience [109,110]. I: implanted/injected, C: chemically induced.

**Table 3 cancers-16-03211-t003:** Studies depicting the role of VDR polymorphisms in various cancers.

S. No.	VDR Polymorphism	Type of Study	Cancer Types	Study Outcome or Key Finding	Reference
1	Apa1 (rs7975232), Cdx2 (rs11568820), and Taq1 (rs731236)	Meta-analysis	23 cancer types	Cdx2 showed increased risk of cancer. Taq1 was associated with increased risk of CRC. Apa1 was not associated with cancer risk.	[229]
2	Apa1 (rs7975232), Bsm1 (rs1544410), Bgl1 (rs739837), and Fok1 (rs2228570)	Nested case–control study and meta-analysis	Keratinocyte cancers	VDR polymorphisms may be associated with the risk of keratinocyte cancers.	[230]
3	ApaI1 (rs7975232) and Cdx2 (rs11568820)	Meta-analysis	Prostate cancer	VDR Cdx2 and Apa1 polymorphisms were not associated with prostate cancer.	[231]
4	Fok1 (rs10735810)	Meta-analysis	Prostate cancer	VDR Fok1 polymorphism could be a promising target and might be capable of causing prostate cancer risk.	[232]
5	Fok1 (rs10735810)	Meta-analysis	Prostate cancer	VDR Fok1 polymorphism may contribute to the risk of developing prostate cancer in Caucasian and population-based studies.	[233]
6	Apa1 (rs7975232), Bsm1 (rs1544410), Fok1 (rs2228570), and Taq1 (rs731236)	Meta-analysis	Breast cancer	VDR Fok1, Bsm1, Taq1, and Apa1 polymorphisms were not associated with the risk of breast cancer in the general as well as Caucasian population.	[234]
7	Fok1 (rs10735810)	Meta-analysis	Sex- and non-sex-associated cancers	Fok1 polymorphism was associated with breast and ovarian cancers.	[235]
8	Bsm1 (rs1544410) and Taq1 (rs731236)	Meta-analysis	Prostate cancer	Taq1 was significantly associated with risk of prostate cancer in Asians and African Americans but not Bsm 1 polymorphism.	[236]
9	Apa1, Bsm1, BgI1, Cdx2, Fok1, Taq1, and Poly (A)	Systematic review and meta-analysis	Breast cancer	VDR gene polymorphisms (Bsm1, Apa1, Fok1, and Poly (A)) may increase susceptibility to breast cancer development.	[237]
10	Taq1	Meta-analysis	CRC	There was no correlation between Taq1 polymorphisms and susceptibility to CRC.	[238]
11	Taq1	Systematic meta-analysis	Prostate cancer	The VDR Taq1 polymorphism might be associated with risk of prostate cancer in Asian (especially Japanese) populations.	[239]
12	Apa1 (rs7975232), Bsm1 (rs1544410), Fok1 (rs10735810), and Taq1 (rs731236)	Meta-analysis	Lung cancer	VDR genetic polymorphism may be correlated with the risk of lung cancer.	[240]
13	Fok1 (rs2228570)	Systematic meta-analysis	CRC	Role of VDR Fok1 polymorphism may differ based on the type and severity of colorectal disease.	[241]
14	Apa1, Fok1, Bsm1, Taq1, and Cdx2	Meta-analysis	CRC	Bsm1 polymorphism was associated with CRC risk, and Fok1 might be a risk factor for CRC.	[242]
15	Fok1 (rs2228570)	Meta-analysis	Ovarian cancer	Fok1 polymorphism increased the risk of ovarian cancer in Caucasian populations in a dominant genetic model.	[243]
16	Bsm1 (rs1544410), Cdx2, and Fok1 (rs2228570)	A systematic review and network meta-analysis	Breast and ovarian cancers	Fok1 and Bsm1 polymorphism are likely the best genetic model for detecting the risk of breast and ovarian cancers, respectively, in Caucasian patients.	[244]
17	Apa1, Fok1, Bsm1, Taq1, and Cdx2	Meta-analysis	Female reproductive cancers	Fok1 and Bsm1 VDR gene polymorphisms may be significantly associated with gynecological cancers.	[245]
18	Apa1 (rs7975232), Fok1 (rs2228570), Bsm1 (rs1544410), and Taq1 (rs731236)	Meta-analysis	RCC	ApaI gene polymorphism and Fok1 FF genotype were associated with RCC susceptibility in Asians.	[246]
19	Apa1 (rs7975232 C > A), Bsm1 (rs1544410 G > A), Cdx2 (rs11568820 T > C), and Taq1 (rs731236 T > C)	Meta-analysis	Lung cancer	Bsm1, Taq1, and Cdx-2 polymorphisms may contribute to lung cancer susceptibility.	[247]
20	Apa1 (rs7975232), Bsm1 (rs1544410), Fok1 (rs10735810), and Taq1 (rs731236)	Meta-analysis	Tobacco-related cancers	Taq1 polymorphism and the risk of tobacco-related cancers were correlated with each other.	[248]
21	A-1012G (rs4516035), Apa1 (rs7975232), Bsm1 (rs1544410), BgI1 (rs739837), Cdx2 (rs11568820), Fok1 (rs2228570), and Taq1 (rs731236)	Systematic review and meta-analysis	Melanoma	Apa1, Bsm1, and Fok1 polymorphisms may influence the development of melanoma.	[249]
22	Apa1 (rs7975232), Bsm1 (rs1544410, A/G), Cdx2 (rs11568820, C/T), Fok1 (rs2228570, T/C), and Taq1 (rs731236, T/C)	Systematic review and meta-analysis	Lung cancer	Bsm1 and Cdx2 polymorphisms decreased lung cancer risk, while Taq1 increased it.	[250]
23	Apa1, Bsm1, Cdx2, Fok1, and Taq1	Systematic review and meta-analysis	18 cancer types	Significant associations with VDR polymorphisms have been reported for prostate (Fok1, Bsm1, Taq1, Apa1, and Cdx2), breast (Fok1, Bsm1, Taq1, Apa1, and Cdx2), colorectal (Fok1, Bsm1, Taq1, and Apa1), and skin cancer (Fok1, Bsm1, and Taq1).	[251]
24	Apa1 (rs7975232), Bsm1 (rs1544410), Cdx2 (rs11568820), Fok1 (rs10735810), and Taq1 (rs731236)	Comprehensive meta-analysis	22 cancer types	VDR polymorphisms were linked to cancer susceptibility. Ethnicity may be a modifier of cancer risk, in particular for hormone-dependent cancers.	[252]

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
