# Peer review of "Vitamin D in Cancer Prevention and Treatment: A Review of Epidemiological, Preclinical, and Cellular Studies"

_cancers, 2024, doi:10.3390/cancers16183211_

Round 1
Reviewer 1 Report
Comments and Suggestions for Authors
The review entitled : Vitamin D in Cancer Prevention and Treatment: A Review of Epidemiological, Preclinical and Cellular Studies" aims to show the molecular mechanisms resposible for anticancer preventive and therapeutic actions of vitamin D and its analogues although, the results of clinical trials concerning this issue are inconclusive. This is very on time topic, but the manuscript requires significant improvements. Generally, sections should be reorganized to present consecutive information with clarity, names of vitamin D forms should be unified and language edition is crucial since some terms are confusing. Please see my comments:
1.The expression " Inhibition of human carcinomas" is not accurate, what do authors mean: prevention of tumor development, prevetion of metastatses development, inhibition of cancer cells proliferation in vitro?
2. Ref 11. in the Introduction [Maiti R. Metronomic chemotherapy. Journal of Pharmacology and Pharmacotherapeutics. 2014;5(3):186-92] does not cover the information in the sentence :"Vitamin D treatment has been shown to significantly slow tumor development and boost cancer cells' susceptibility to chemotherapeutic drugs". Moreover, the authors should be more precise regarding presented data, namely did you mean the results of in vitro or animal model studies?.
3. In lines 82-86 the authors used term geographical/ecological studies. To the best of my knowledge clinical trials are divided into observational and intervention studies, and observational studies include epidemiological studies, cross-sectional, case control etc. Among clinical trials there are no geographical or ecological studies. Did you mean that geographic region of living and related sun exposure affect vitamin D level and cancer risk? I do not know what did you mean in case of ecological studies.
4. What is the association between the first sentence of the second pargraph of 1.1 section ("Studies have reported that vitamin D modifies the effects of advanced glycation end- products (AGEs) and their receptor [15]. ") with the rest of the paragraph. This section is unclear, authors should explain the structure and mechanisms of function VDR, next the effect of various single nucletide polymorphisms (SNPs in VDR gene) on VDR function. Again, to understand the effect of SNPs in CYP2R1 gene, on cancer risk related with vitamin D level, the role of CYP2R1 should be provided first.
5. In the next paragraph, the role of CYP24A1, and CYP2R1 should be described.
6. Did you mean confounding factors in line 128 using term "various confirming factors"
7. In the paragraph between line 131 and 145, authors used the followinf terms "calcitriol", 25(OH)D, circulating vitamin D. A the beginnig of manuscript it should be explained the meaning of this terms beacuse 25(OH)D is the same as circulating vitamin D or calcidiol. Whereas, calcitriol is an active form of vitamin abbreviated also as 1,25(OH)2D. Please explain also why for evalution of vitamin D level, 25(OH)D is used instead of 1,25(OH)D?
8. The sentence in lines 144 and 145 "These results support a positive relationship between circulating 25-OHD and better survival in cancer patients [36]." indicates that positive correlation is between higher circulating 25OHD level (>81 nmol/L) and better survival in cancer patients.
9. Please check the description of the results of study from ref . 37 because group were divided into low (<46nmol/L) and high (≥46 nmol/L) categories, but the next sentence does not correspond to categories. It refers to the concentration ≥62 nmol/L
10. Please explain what do you mean using "total circulating vitamin D" in line 156?
11. The description of the results from ref. 39, the classification into groups beased on 25(OH)D level lacks group with 30-50 nmol/L
12. Does gender affect vitamin D level and cancer risk? Please sum up, the paragraph between line 155 and 171.
13. Referring to the results of ref 52., what about the effect of vitamin D supplementation on non invasive cancers risk?
14. Can you comment whether other RCTs apart from ref. 50 or 51 did not overlook the duration of 25(OH)D in the blood?
15. The essential information for section 1.2 is epidemiological data about vitamin D hypervitaminosis frequency.
16. From the section 1.3 one can see that vitamin D and its analogoues exert anticancer action via the same mechanisms, thus there is no need to describe it in two separate paragraphs (248-272) . However, this section lacks "the toxic effects associated with the administration of vitamin D". Please list these effects.
17. The 1.4 section contains not only metabolism, but also mechanisms of vitamin D, thus the title should be changed. Moreover, this section should be moved to the begining of the manuscript to better understand the information presented in 1.1 section
18. Similary, section 3.1 should be presented before 1.1 section
19. When authors write about SNP they should use rs numebr proparticular SNP instead of names of endonucleases that recognize SNP (eg Fok1, Apa1). This is especially confusing in case of CDX2 which is the name of gene encoding transcription factor, thus information in line 549-553 should be presented clearly.
20. What are examples of long-term toxicity of vitamin D mentioned in section 4
21. The chapter 4 involving vitamin D analogues should be before chapter 2, since their molecular action were presented in chapter 2.
22. In table 22, the title of column cancer type is confusing, eg what is the difference between "CRC xenograft" and "Mice were inoculated with CRC cells", "Chemically induced CRC". Please add more information such the name of animal used, where the cancer cells were inoculated, what chemical compound was used to induce CRC. Moreover, additional column with obtained effect of vitamin D should be introduced.
23. In the Abstract there is the following sentence: "The existing empirical data has demonstrated that vitamin D can also work in the absence of vitamin D receptor (VDR) indicating the presence of multiple mechanisms of action for this sunshine vitamin."Please indicate in which part of the manuscript you addressed to this sentence
Comments on the Quality of English LanguageExtensive editing of English language required.
Author Response
The authors would like to thank the reviewer for thoroughly reviewing the manuscript. Please find the following attachment for a point-by-point response.

Reviewer 2 Report
Comments and Suggestions for Authors
Dallavalasa et al made a comprehensive review von vitamin D and cancer prevention
When citing and commenting on different papers where nmol or mg/dl were used, always state what the authors of the original paper used. For a comprehensible summary, you should take the trouble to convert this into a standardised unit, in my opinion mg/dl. You could announce this in the introduction.
Page 2 second paragraph (line 91-107): after the description of the basic association of vitamins and cancer prevention, the authors go directly to the VDR and pathomechanisms, in my opinion this is a different chapter and should either be included under chapter 3 or have its own heading here, from page 3 107 it continues in the correct style
page 4 171 may have its own sub-chapter: Supplementation trials ?
page 5 214 : one main critic of the VITAL trial is that baseline levels are too high to show a benefit of supplementation may be added.
Table 1 should be higher up where it is mentioned
line 270 typo : DVR instead of VDR
Chapter 3.2. VDR is not easy to digest and understand, can you give a table on polymorphisms, cancer type, relative risk and availability of validation/confirmation cohorts? I think the chart you produced does not contain enough information
Comments on the Quality of English Language
Quality of english is sufficient however on some chapters especially 3.2. the readibility could be improved.
Author Response
The authors would like to thank the reviewer for thoroughly reviewing the manuscript. Please find the following attachment for a point-by-point response to the comments.

Round 2
Reviewer 1 Report
Comments and Suggestions for Authors
Dear Authors,
you have responded sufficiently to all my comments and now the manuscript is cleare,
best regards,
Reviewer 2 Report
Comments and Suggestions for Authors
Dallavalasa et al made a comprehensive review von vitamin D and cancer prevention, it contains a good oversight on current knowledge from clinical trials as wells as basic research, especially the review of vitamin D analogues is a very useful reference for future research.